# Executive Functions as Measured by the Dog Executive Function Scale (DEFS) over the Lifespan of Dogs

**DOI:** 10.3390/ani13030533

**Published:** 2023-02-02

**Authors:** Maike Foraita, Tiffani Howell, Pauleen Bennett

**Affiliations:** Anthrozoology Research Group, School of Psychology and Public Health, La Trobe University, Melbourne 3086, Australia

**Keywords:** dog cognition, dog behaviour, behavioural regulation, executive functions, lifespan development, working dogs

## Abstract

**Simple Summary:**

Dogs need to effortfully self-regulate their behaviours using cognitive skills such as inhibition or working memory. These skills are called Executive Functions (EF) and can be assessed through behavioural research, or owner-rated questionnaires about the dog’s behaviour. This study investigated whether the Dog Executive Function Scale developed for adult dogs can be used in young (<1 year) and old (>8 years) dogs. Results suggest that young, adult and old dogs’ EF can all be assessed by the same scale, as they have the same components. The lifespan development of these cognitive skills varied. Working memory and a measure of attention showed an increase in the first years and a decline thereafter. Different forms of inhibition showed complex associations with age (e.g., steady increase or steady decrease). Dogs that received more training, and working dogs, had better EF, independent of age. Training history appeared more important for EF in non-working dogs than working dogs, perhaps because all working dogs receive a high level of training. Pet owners wanting to improve their dogs’ behavioural regulation can be encouraged to partake in a variety of training activities with their dogs, independent of their dog’s age.

**Abstract:**

Executive Functions (EFs) are needed for effortful self-regulation of behaviour and are known to change over the lifespan in humans. In domestic dogs, EFs can be assessed through behavioural rating scales, such as the Dog Executive Function Scale (DEFS). The primary aim of this study was to investigate whether the DEFS, developed initially using a sample of adult dogs, can be used in juvenile (<1 year) and senior (>8 years) dogs. Confirmatory factor analysis of a juvenile and senior dog sample led to good model fit indices, indicating that juvenile and senior dogs’ EF structure follows the same functional organisation as found in the DEFS. The secondary aim was to analyse the lifespan development of EFs. Analysis of subscale scores revealed multifaceted relationships with age for four subscales. Working Memory and Attention Towards Owner showed the u-shaped curve traditionally associated with the lifespan development of EFs. Forms of inhibition showed complex associations with age, i.e., Delay Inhibition declined in aging and Motor Regulation increased during aging. Training history and Working Status influenced performance independent of age. More highly trained dogs and working dogs exhibited higher EF skills. Training history appeared more important for EF in non-working dogs than working dogs, perhaps because all working dogs receive a high level of training.

## 1. Introduction

Executive Functions (EFs) are a set of cognitive processes that are important for behavioural self-regulation. Inhibition, working memory, cognitive flexibility and attention are skills commonly referred to in the literature as EFs [1,2]. These, and others, make up a complex group of different EF skills that are used to regulate behaviour in a deliberate way. Many dogs (*Canis lupus familiaris)* live and work alongside humans throughout their lifespan, and they most likely use EFs to effortfully regulate their behaviour. Both pet and working dogs need to inhibit urges to chew or dig, and they need to memorise cues and ignore distractions. In fact, a growing body of research is demonstrating the importance of cognitive skills and EFs for working success in working dogs [3,4,5,6,7,8,9,10,11].

There are two features of EF that are dynamic. The first is the functional organisation of EF, i.e., the number of distinguishable components. In humans, the structure of EFs is complex and appears to follow a concept called “unity and diversity”, with EF components being partially separable and partially correlated, which has led to various hypotheses about the underlying structure of latent factors [12,13]. The unity part of EF might be explained by correlations among latent EF factors, or by the existence of a central EF factor that influences other EF factors [12]. Which different components of EFs can be identified appears to be age dependent. EF skills seem to reflect a more unified construct early in life, which then separates into more diverse skills during development [14,15]. A commonly used behavioural rating scale called the Behavior Rating Inventory of Executive Function (BRIEF) can be used to measure this in humans. It was first developed for school-aged children [16], but then adapted for pre-schoolers [17] and for adults [18]. The number of distinct EF factors that are identified in these cohorts varies, with five in pre-schoolers, 8 in school-aged children and 9 in adults.

Given the similarities between human and canine cognition that have been found across multiple domains, such as cognitive aging [19] and social cognition [20] we might expect a similar development in dogs: fewer separable factors in young dogs, with an increase in structural complexity in adult dogs. However, EFs that emerge later in human development are typically higher order EFs, such as Self-Monitoring (i.e., ability to monitor effects of behaviour on others) and Organisation of Materials, [i.e. keep personal environment in an orderly manner, [21]. Higher-order EFs might build upon EFs that develop earlier and may be related to the maturation of the prefrontal cortex in humans [14]. If so, dogs’ functional organisation of EFs might lack some of the higher-order skills observed in adult humans, and maturation of EF structure therefore might occur quite early in development. To our knowledge, no studies have aimed to compare EFs functional structure across different age groups in dogs.

The second dynamic feature of EF is performance on those distinguishable components. In addition to the impacts of age on EF structure in humans, EF abilities are also associated with age. Scores on various EF measures change throughout development. EFs have been found to generally increase during early life, peak during middle age, and decrease during aging [22,23]. However, there is also some evidence for age invariance from adulthood to old age for certain EF skills, such as some aspects of inhibitory control [24], and even increases in performance from adulthood to old age, such as executive attention [25]. In dogs, studies have demonstrated an increase in cognitive capabilities in early life. This has mainly been shown in working dogs [26,27]. Additionally, a myriad of studies has demonstrated a decline in EF skills during aging in pet and laboratory dogs [19,28,29,30,31,32,33,34,35]. There is evidence that training might mitigate cognitive decline during aging [30], but few studies have attempted to follow the trajectory of cognitive skills throughout the whole lifespan in dogs. A recent study [36] demonstrated that a broad set of cognitive skills follow an inverted u-shaped trajectory in pet dogs. Amongst these were memory, measured with a delayed choice task, and delay of gratification, measured as the ability to follow an instruction by the owner to leave a treat. Given the complex trajectories of EF skills found in humans, more research is needed that follows the lifetime trajectory of various EFs and other cognitive skills in dogs.

One way of assessing EF abilities in humans is through self-reported or parent/teacher behavioural rating scales [17,37,38,39]. There is evidence that EF skills can be assessed via owner-reported behavioural scales in dogs also [40,41,42]. We developed a behavioural rating scale aimed at measuring multiple components of EF in adult dogs [40]. Six different factors of EF were identified that partly match EF factors previously identified in humans and describe different cognitive skills needed in a variety of situations. Behavioural Flexibility describes the dog’s ability to adapt to new situations (e.g., “My dog gets upset about changes in the environment (e.g., a new piece of furniture)”). Motor Regulation describes the dog’s ability to control and inhibit motor functions during high arousal situations (e.g., “My dog needs constant reminding to control behaviours which are inappropriate (e.g., jumping up on visitors)”). Delay Inhibition describes the dog’s ability to control behaviour when anticipating something desirable (e.g., “My dog finds it difficult to tolerate waiting for a reward”). Attention Towards Owner describes how well the dog pays attention towards its owner (e.g., “My dog gazes at me or turns toward me when I speak to him/her”). Instruction Following describes how dogs follow instructions in various situations (e.g., “My dog will follow instructions (e.g., ‘sit’ or ‘stay’) given by a stranger”). Lastly, Working Memory describes the dog’s concentration on tasks and ability to keep objects/activities in mind when they are no longer perceptually present (e.g., “My dog forgets about something he/she wanted once it is out of sight (e.g., toy, food)”). To our knowledge, the DEFS is the first study to characterise a factor structure of EF in dogs, but whether the complexity of EF structure varies over a dog’s lifespan remains to be investigated.

The primary aim of this study was to evaluate whether the Dog Executive Function Scale (DEFS) for adult dogs, with its 6-factor structure, can be used to measure EFs in juvenile and senior dogs, thereby allowing comparisons of EF structure across age groups in dogs. If the DEFS is suitable to be used on juvenile and senior dogs, a secondary aim was to evaluate the trajectory of the 6 subscales (i.e., Behavioural Flexibility, Motor Regulation, Delay Inhibition, Working Memory, Instruction Following and Attention Towards Owner) over dogs’ lifespan and across training history.

## 2. Materials and Methods

A survey with items aimed at measuring EF components was distributed to a convenience sample of dog owners [40]. Included in this survey were the items that make up the DEFS [40]. Confirmatory factor analysis (CFA) was used to determine whether the DEFS for adult dogs [40] captures EF structure in juvenile (<1 year) and senior (>8 years) dogs. Dogs’ scale scores were then compared across the lifespan from youth to senescence, and across training background using generalised additive modelling. This study was approved by the La Trobe University Science, Health and Engineering College Human Ethics Sub-Committee, (approval number: HEC19533).

An online survey was distributed, containing the 23 DEFS items as part of a larger survey [40] together with demographic questions about the participant and their dog. These included the dog’s sex and reproductive status, whether the dog was a working dog/trainee working dog or not, and the dog’s training history, from which a Training Score was calculated by summing different types of training the dog had received throughout its life. Both fully qualified and trainee working dogs were classed as working dogs for the analysis, so that juvenile dogs could be classed as working dogs as well. Participants were required to be at least 18 years of age and take care of at least one dog. A 5-point Likert scale ranging from “Never or almost never” to “Always or almost always” was used to administer survey items, with a “not applicable” option where needed. Invitations to take part were distributed worldwide via social media. For full details on the survey’s development and distribution, as well as the full survey and scoring instructions, see [40]

Item responses were reverse coded, if necessary [see Dog Executive Function Scale – Coding Key, 40], with high numbers indicating high EF capabilities and low numbers indicating low EF capabilities. The sample was split into groups by dog’s age (juveniles: <1 year, adults: 1–8 years, seniors: >8 years). The juvenile and senior samples were used for confirmatory factor analysis. Likert scale surveys create ordinal data, for which factor analysis based on polychoric correlations is prudent [43]. Polychoric confirmatory factor analysis was performed on the juvenile dog (*n* = 129) and senior dog (*n* = 127) samples using lavaan version 0.6–8 in R version 4.0.0, with pairwise deletion of missing values, using the 6-factor model structure from the DEFS. In accordance with the DEFS model [40], covariances between latent variables were permitted and diagonally weighted least squares estimation was used. The 6-factor DEFS model was compared to a model with a single latent factor and an orthogonal model with 6 latent variables. CFA was performed in R version 4.0.0, using the function “cfa” from the package “lavaan” with pairwise deletion of missing values.

Model fit was estimated using the model chi-square (χ^2^), the root mean square error of approximation (RSMEA), the comparative fit index (CFI), the Tucker-Lewis index (TLI) and the standardised root mean square residual (SRMR, Table 1). Better model fit is indicated by lower χ^2^ values in relation to the degrees of freedom, together with a non-significant χ^2^
*p*-value [44]. The RSMEA and SRMR are so-called “badness of fit” measures, with a value close to zero indicating good fit. Values below 0.06 (RMSEA) and 0.08 (SRMR) are often referred to as indicating good fit [44]. Higher values indicate better model fit in the CFI and TLI, with values at or above 0.95 considered to show good model fit [44]. 

The complete sample was used for lifespan comparisons of EF scores. Calculation of subscale scores for each participant was performed by summing each item score and dividing by the number of items (see Dog Executive Function Scale—Coding Key, [40]). Cognitive skills have been shown to follow a variety of different shaped trajectories over the lifespan, such as linear or inverse u-shaped trajectories. To investigate trajectories, generalised additive models (GAMs) were fit. GAMs are an extension of linear regression models that do not assume normality of the data and use smoothing splines that capture non-linear responses to predictor variables very efficiently [45]. Gamma distributed GAMs with logit link functions, using the subscale scores as response variables and age in years, Training Score, Working Status and sex as predictors were fit. 

Age in years and Working Status might have an impact on the dog’s Training Score. Older dogs have had more time to receive different kinds of training, and, while working dogs are most likely highly trained, they might not score highly on a measure of training based on the summation of different types of training received. Therefore, interaction terms of training score and Working Status, as well as Training Score and year of age, were included in the models. Penalised thin-plate regression splines, which allow for the whole term to be shrunk to zero, were used for smoothing the predictors, except for sex and Working Status, which were added as factors. Where the model identified significant differences between factor levels with more than two levels, pairwise Wilcoxon tests were performed with the correction of *p*-values according to Benjamin and Hochberg [46]. The function “gam” from the packaged “mcgv” in R version 4.0.0. with a “gamma” distribution and “log” link and method “REML” were used for all models. The models looked as follows:Subscale Score ~ s(Age in Year)+s(Training Score)+ti(Age in Years, Training Score)+s(Training Score, by=Working Status)+Working Status+Sex

## 3. Results

In total, 1239 participants took part in the survey, with 1066 providing age data for their dog. Of these, 164 responses belonged to dogs under 1 year of age (juveniles), 147 belonged to dogs over 8 years of age (seniors) and 755 belonged to dogs between 1 and 8 years of age (adults). Participants with more than 10% missing responses and who did not supply the Working Status of their dog were excluded. This left 129 participants in the juvenile sample, 127 participants in the senior sample, and 698 participants in the adult dog sample. 

Juvenile dogs were aged 3–11 months (M = 7.58 months, Median = 7.5 months, SD = 2.36 months), adult dogs were aged 1–8 years (M = 3.9 years, Median = 4 years, SD = 2.25 years) and senior dogs were aged 9–15 years (M = 11.81 years, Median = 12 years, SD = 1.43 years). The juvenile dog sample consisted of 33.3% intact females, 14.0% spayed females, 35.7% intact males and 17.0% desexed males. Of the adult dogs 7.2% were intact females, 38.3% were spayed females, 12.6% were intact males and 41.9% were desexed males. The senior dog sample consisted of 4.7% intact females, 50.4% spayed females, 4.7% intact males and 40.2% desexed males.

Juvenile dog owners were aged 19–70 years (M = 39.37 years, SD = 13.58 years) and were 88.1% female. Most participants were born in either Australia (29.6%), the UK (21.5%), or the USA (20.7%), with the remainder of participants being born in various other countries. Adult dog owners were aged 18-76 years (M = 37.21 years, SD = 13.02 years), and 87.6% were female. The highest percentage of participants were born in Australia (41.5%), followed by the USA (16.3%) and the UK (13.8%). Senior dog owners were aged 18-78 years (M = 42.79 years, SD = 13.92 years) and were 83.4% female. Most participants were born in Australia (33.1%), the UK (13.4%), and the USA (11.0%).

### 3.1. Confirmatory Factor Analysis

There are varying recommendations for sample size requirements in factor analysis. One such rule suggests a sample size of 3–6 times the number of variables [47], which equates to 69 to 138 participants for the 23 variables in the DEFS. Another states a minimum of 100 data points [48] We, therefore, went ahead with the analysis. For the original 6-factor DEFS model, applied to the juvenile and senior dog data, all but one parameter was within values that indicate good model fit (Table 1). The SRMR for both juvenile and senior dogs was 0.08, which is just at the threshold value for a good fit. Additionally, both juvenile and senior dog data were significantly better modelled by the original model than an orthogonal model with 6 latent variables (juvenile dog: *p* < 0.001 ***; senior dog: *p* < 0.001 ***) or a model with a single latent factor for executive function (juvenile dog: *p* < 0.001 ***; senior dog: *p* < 0.001 ***). Standardised and unstandardised regression coefficients for the 6-factor solution, permitting covariance between the factors, are shown in Table 2. Overall, juvenile and senior dog EF structure can be well-modelled by the 6-factor solution found in adult dogs, indicating that comparison of scale scores over the lifespan from 0 to 15 years is appropriate.

### 3.2. Life Span Trajectories of Subscale Scores and Other Predictors

Full model statistics for all subscales are visible in Table 3; significant relationships are indicated by bold *p*-values. Age in years was a significant predictor of four subscale scores, namely Motor Regulation, Delay Inhibition, Working Memory and Attention Towards Owner, while there was no relationship between age in years and Behavioural Flexibility or Instruction Following (Table 3). The smoothed predicted values of the four subscale scores with a significant relationship to age in years, show varying trajectories across the lifespan (Figure 1). Motor Regulation increases steeply in the first 3–4 years, with a less steep increase thereafter. Delay Inhibition steadily decreases over the lifespan. Working Memory shows an inverted u-shape, with an increase until around 8 years and a decline thereafter. Attention Towards Owner increases sharply over the first 4 years, with a more gradual decline thereafter. There was no interaction effect between age in years and Training Score for any of the subscales.

Working Status had a significant relationship with all subscale scores, with working dogs scoring higher than non-working dogs on all subscales (Table 3). Training Score was a significant predictor for Behavioural Flexibility and Attention Towards Owner. Behavioural Flexibility and Attention Towards Owner scores increase with higher Training Score. There was a significant interaction between Training Score and Working Status in Motor Regulation, Delay Inhibition, Working Memory, Instruction Following and Attention Towards Owner. In these scale scores, training history affects working and non-working dogs differently (Figure 2). Generally, the subscale score is more strongly affected by Training Score in non-working dogs, while Training Score has a smaller effect on the subscale score in working dogs (Figure 2).

GAMs indicated significant relationships between sex and reproductive status with the subscale scores Working Memory, Behavioural Flexibility and Motor Regulation. To investigate group differences, pairwise Wilcoxon tests were performed with the correction of *p*-values according to Benjamin and Hochberg [46] (see Table 4). Intact females (mean = 4.04, SD = 0.85) and intact males (mean = 3.98, SD = 0.79) scored significantly higher than desexed males (mean = 3.74, SD = 0.86) in Behavioural Flexibility. Intact (mean = 2.69, SD = 0.89) and desexed (mean = 2.67, SD = 0.87) males scored significantly lower than spayed females (mean = 2.94, SD = 0.89) in Motor Regulation. Intact males (mean = 3.63, SD = 0.65) scored higher than desexed males (mean = 3.43, SD = 0.70) in Working Memory.

## 4. Discussion

The primary aim of this study was to determine the validity of the Dog Executive Function Scale (DEFS), developed in adult dogs, for samples of juvenile (<1 year) and senior (>8 years) dogs. Responses from owners of 129 juvenile and 127 senior dogs were analysed. Confirmatory factor analysis of both the juvenile and senior samples led to good model fit indices (Table 1), indicating that juvenile and senior dogs’ executive function (EF) structure follows the same functional organisation as adult dogs’ EF structure. Juvenile and senior dogs’ EF can be well-modelled by the 6-factor solution, with the factors representing Behavioural Flexibility, Motor Regulation, Delay Inhibition, Working Memory, Instruction Following and Attention Towards Owner.

In our study, the 6 factors of EF that had previously been identified in an adult sample proved stable in a confirmatory factor analysis in young dogs under 1 year of age. Our sample of young dogs’ ages ranged from 3 to 11 months, with a mean of almost 8 months. Dogs develop rapidly during early life stages [49,50] and cognitive traits have been shown to improve during the first 12 months of life [27,51]. It appears that at the age of our sample, dogs’ EF structure had reached adult complexity. Unfortunately, to investigate this functional organisation in greater temporal detail, owner-rated behaviour scales are not a feasible tool. In many countries, puppies are expected to be at least 8 weeks of age before they leave their mother and litter [52]. This means that breeders would need to complete behaviour scales, rather than owners, and they might only spend limited time with individual puppies and not develop a good understanding of individual puppy capabilities. Studies have shown that common EF skills such as inhibition and working memory can be assessed in dogs as young as 7.5–8 weeks of age through laboratory tasks [53,54]. To investigate the factor structure of young puppies, therefore, test batteries of various cognitive tests might be able to be employed, and exploratory and confirmatory factor analysis used to identify separable domains. Undertaking this task was beyond the scope of the current study. 

Given the decline in EF [36] and general cognitive abilities found in dogs during aging, it might be expected that factor structure would change in aged dogs as well, when compared with an adult sample. We found no evidence of this, with EF factor structure appearing not to be influenced by age. Overall, our results suggest that the Dog Executive Function Scale can be used with juvenile, adult as well as senior dogs.

Given the findings that the DEFS can be used across a wide age span in dogs, the secondary aim was to investigate lifespan trajectories of, and influence of training history on, EF factors from youth to senescence. In humans, EF have been shown to increase from infancy to early adulthood, and then plateau or decline [22] and in some instances increase [25] during aging. In our sample of dogs, age in years was a significant predictor of Motor Regulation, Delay Inhibition, Working Memory and Attention Towards Owner, with varying developmental trajectories.

Working Memory and Attention Towards Owner showed an inverted u-shaped trajectory across the lifespan. This trajectory, of marked increases during early years and a decline in age, has traditionally been associated with EF overall [22]. Working memory’s trajectory over the lifespan in humans is an inverted u-shaped curve [55], and a u-shaped trajectory over the lifespan in working memory has been observed in dogs [36]. Working Memory in dogs, according to the DEFS, follows a similar trajectory, with an increase from 0 to about 8 years, and a decline thereafter. Attention has been shown to decrease in dogs with age [30,31,56]. Social attention towards humans increases over the first two years of life [26]. Eye contact, a measure of social engagement, has been found to exhibit an inverse u-shaped trajectory also [36]. In our data, Attention Towards Owner appears to increase steeply from 0 to 4 years, plateaus for a year and then declines until 15 years (Figure 1). While Attention Towards Owner is quite a narrow part of attention, it is likely the easiest for dog owners to recognise in owner-rated EF assessments. Attention Towards Owner follows the same trajectory as other domains of attention observed in dogs.

More recently, human EF research has demonstrated that not all components of EF decline during aging [57]. Some remain stable in old age [24], while others even show increases in age [25]. Our results mirror this diverse development of EF, with Delay Inhibition decreasing throughout life, and Motor Regulation increasing throughout life. Both factors are likely to reflect different domains of inhibition. Motor Regulation is related to the control of motor patterns in situations of high arousal and is a form of motor inhibition. Delay Inhibition describes the dog’s ability to control behaviour when waiting for something highly anticipated. Traditionally, inhibition in humans has been found to decline during aging, a phenomenon called the inhibitory deficit hypothesis [58,59,60]. However, different forms of inhibition appear to follow varying trajectories. A meta-analysis demonstrated that some forms of inhibition are age-invariant from adulthood to old age [24], or even continue to increase during aging [25].

It is likely that overall reduced motor activity with age [61] contributes to the lifelong increase seen in Motor Regulation. Laboratory cognitive measures that involve little motor activity (i.e., choosing between competing stimuli on a screen), and therefore facilitate differentiation between motor activity and motor regulation, should be used to further investigate aging of inhibition in dogs.

Surprisingly, we found no increase in Delay Inhibition during the early years in dogs. Inhibition is one of the first EFs to develop in humans [14]; dogs develop rapidly during early life stages and inhibition has been measured in puppies as young as 8–10 weeks [53,54]. It is possible that dogs reach peak performance in Delay Inhibition before finishing their first year of life. An increase in performance therefore may not have been detected with the temporal resolution used in our study.

Two subscales, Behavioural Flexibility and Instruction Following, showed age invariance across the lifespan. There are multiple possible explanations for this, one of which might be the temporal resolution used, which might not have detected increases happening early in life. However, it is also possible that other factors that are age-independent influence Behavioural Flexibility and Instruction Following. Such factors could be personality traits for instance. Personality and cognitive traits can be difficult to disentangle [62]. Investigating all influencing factors was outside the scope of this study, and future research might investigate further influences on Behavioural Flexibility and Instruction Following.

Training and practice are known to affect EF performance positively in humans [63,64] and dogs [56,65]. Part of the data used in this study, specifically the adult dog subset with dogs aged 1-8 years, was already analysed for demographic comparisons of subscale scores with Training Score and Working Status [40]. However, we included both Training Score and Working Status in our models here to check for confounding effects with age, and because the complete set of dogs aged 0 to 15 years had not been analysed for effects of training. Congruent with these findings, all DEFS subscales were significantly influenced by either the Training Score, Working Status, or the interaction of Training Score and Working Status. Training requires dogs to pay attention, to memorise cues and to ignore distractions. Working dogs can be expected to receive large amounts of well-structured training to prepare them for their working role. Training leads to improvements in a dog’s effortful self-regulation of behaviour. Interestingly, no interaction between the dogs’ age and Training Score was detected, although this might also reflect the limited temporal resolution in the current study. Training appears to be beneficial for EFs independent of the dogs’ age. 

For Motor Regulation, Delay Inhibition, Working Memory, Instruction Following and Attention Towards Owner, the effect of the Training Score on the subscale score is different for working dogs and non-working dogs (Figure 2). While non-working dogs’ EF scores benefit from more types of training received, this effect is less pronounced for working dogs. This might reflect a ceiling effect in working dogs, which typically receive a large amount of structured training. Any additional types of training they receive may have a smaller or no effect on their EF. This means working dog organisations are most likely already providing sufficient training to challenge and maximise their dogs EF capabilities, although it is also likely that these organisations select or breed dogs based on their having strong EF functions.

Sex and reproduction status differences were found in Working Memory, Behavioural Flexibility and Motor Regulation. Males did score lower than spayed but not intact females in Motor Regulation, a form of motor inhibition. Males have been found to be more impulsive [66]. In our study, intact males scored higher than desexed males, but not higher than females, in Working Memory. Male dogs have been found to perform slightly better in a delayed memory task [36], but female dogs have been found to show better spatial memory [67]. More studies are needed to clarify the direction, mechanisms and causes for the difference in cognition across sex and reproduction status in dogs. 

There are some limitations of the study to consider. Given the rapid behavioural, social and cognitive development observed in dogs in their first weeks of life [49], it is likely that EFs develop rapidly in this time period as well. Indeed, cognitive traits improve during the first 12 months of life [27,51] and EFs such as inhibition can be assessed in puppies as young as 7–8 weeks [53,54]. The temporal resolution used in this study is years, and the sample of juvenile dogs’ mean age was 7.5 months. Failure to detect changes in EF structure in young dogs could be due to this lack of temporal detail. Future studies should aim to look at the development of EF structure in more temporal detail.

The Training Score used in this study to assess dogs’ training history is an approximation. The highly variable nature of the training received by dogs makes quantifying training history impossible, particularly in a survey study. The Training Score, calculated by summing different types of training received, used in this study makes training history accessible for analysis. Similar methods have been used successfully [30,56] to reveal associations between training history and cognitive characteristics. While Training Score might not be a good measure to assess an individual dog’s level of training, this approximation can be expected to represent differences on a population level, e.g., on average a dog that only experienced one type of training (e.g., trick training at home) spent less time training than a dog that experienced 8 different types of training. Owners were asked whether their dog is a working dog/trainee working dog, but no definition of working dog was provided. While we expect the majority of the sampled population to understand working dogs to be the common working dogs such as assistance dogs, therapy dogs or sniffer dogs, it is possible that some people did not. Finally, the data used in this study is cross-sectional (i.e., each dog is described by its owner at one age, no longitudinal data for individual dogs were collected). While this cross-sectional data can give valuable insights into development on a population level, future research might use longitudinal designs to follow the trajectory of EFs in individual dogs.

The Dog Executive Function Scale (DEFS) can be used to assess components of EF in juvenile to senior dogs. Researchers can use the DEFS in this age range in dogs together with other cognitive measures, to further validate and refine the scale. A higher Training Score is associated with better behavioural self-regulation in dogs, with no interaction between dog age and Training Score. Pet owners seeking to improve their dogs’ behavioural regulation can be encouraged to partake in a variety of training activities with their dogs, independent of their dog’s age. Working dog organisations are known to select dogs carefully, but appear nonetheless to expose their dogs to sufficient training to challenge and maximise their dogs’ abilities to behaviourally self-regulate.

## 5. Conclusions

The primary aim of this study was to investigate the validity of the Dog Executive Function Scale (DEFS) for juvenile and senior dogs. The results indicate that functional organisation of executive functions (EF) is stable over dogs’ lifespan and that the DEFS can therefore be used for juvenile to senior dogs. Resulting from this, we could analyse the lifespan development of EFs in dogs. Four of the six subscales are age-dependent, and the development of the different scales is multifaceted. U-shaped-curves traditionally associated with the development of EF were observed for Working Memory and Attention Towards Owner. Different forms of inhibition seem to be differentially affected by age, with a steady increase in Motor Regulation and a steady decline in Delay Inhibition. A limited temporal resolution might have played a role in failing to detect increases during early life in Motor Regulation and Delay Inhibition. Training positively impacts dogs’ EF, independent of age, but more so in non-working dogs. Working dogs appear to receive sufficient training to maximise EF and behavioural self-regulation.

## Figures and Tables

**Figure 1 animals-13-00533-f001:**
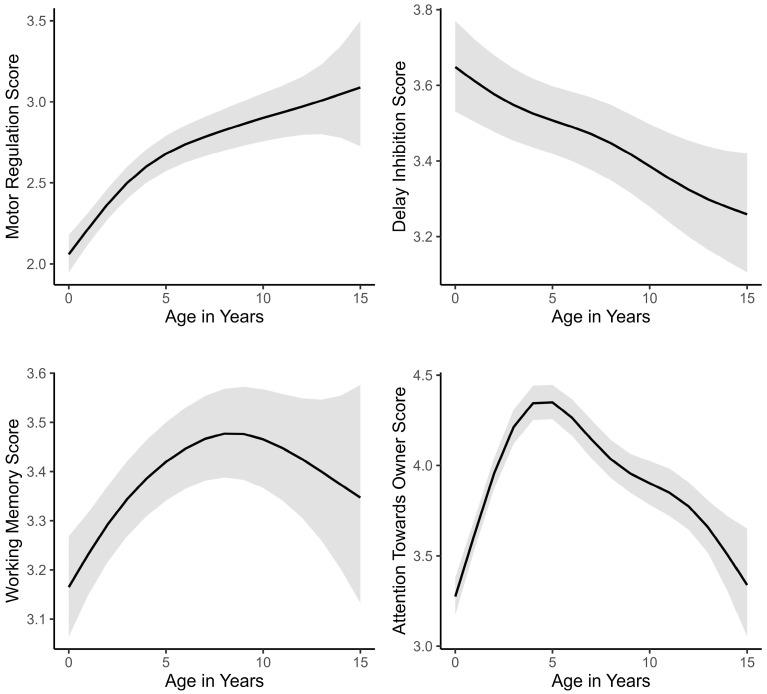
Estimated smoothing curves of the relationship of subscale scores for Motor Regulation, Delay Inhibition, Working Memory and Attention Towards Owner and dogs’ age in years. Solid lines are the smoothers and shaded regions indicate 95% confidence intervals.

**Figure 2 animals-13-00533-f002:**
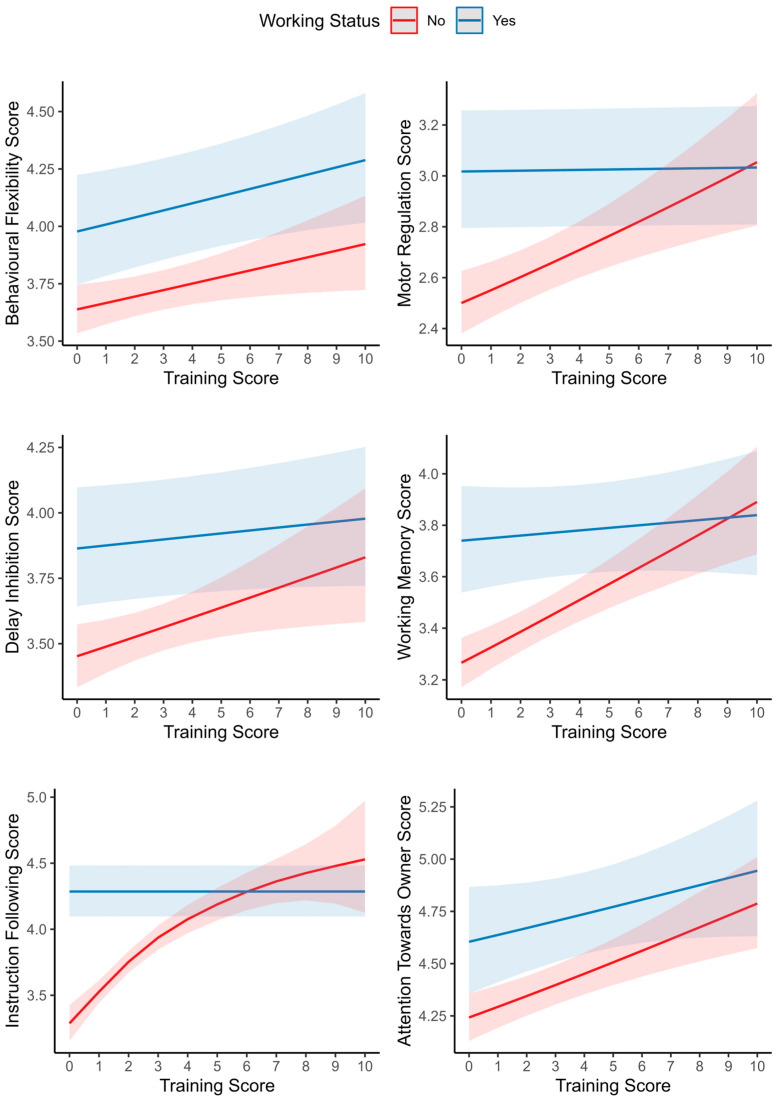
Estimated smoothing curves of the relationship of subscale scores and dogs’ Training Score by Working Status. Solid lines are the smoothers and shaded regions indicate 95% confidence intervals.

**Table 1 animals-13-00533-t001:** Fit indices for models using juvenile and senior dog data. DEFS Model = Oblique model with 6 latent factors, Orthogonal Model = Orthogonal model with 6 latent factors, Single Model = Model with single latent factor for executive function. Values that indicate good model fit are bold.

	Juvenile Dog Sample (*n* = 129)			Senior Dog Sample (*n* = 127)		
	DEFS	Orthogonal	Single	DEFS	Orthogonal	Single
df	215.00	230.00	230.00	215.00	230.00	230.00
χ^2^	238.45	1317.31	750.64	243.09	1289.53	1035.07
(χ^2^) *p*	**0.13**	0.00	0.00	**0.09**	0.00	0.00
RSMEA	**0.03**	0.19	0.13	**0.03**	0.19	0.17
CFI	**0.99**	0.65	0.83	**0.99**	0.71	0.78
TLI	**0.99**	0.61	0.81	**0.99**	0.68	0.75
SRMR	0.08	0.18	0.12	0.08	0.18	0.15

**Table 2 animals-13-00533-t002:** Standardised and unstandardised regression coefficients for the oblique 6-factor solution (DEFS model). *p*-values for all latent variables are < 0.001 ***, with the following exceptions: WM4 juvenile *p* = 0.056, WM4 senior *p* = 0.883, IF4 juvenile *p* = 0.003.

Latent Variable	Indicator	B (Unstandardised Estimates)		β (Standardised Estimates)		SE		Z	
		Juvenile	Senior	Juvenile	Senior	Juvenile	Senior	Juvenile	Senior
Behavioural Flexibility	BF1	1.000	1.00	0.510	0.422	0.000	0.000		
	BF2	1.651	2.128	0.842	0.897	0.292	0.501	5.645	4.246
	BF3	1.037	1.676	0.529	0.706	0.205	0.379	5.056	4.427
	BF4	1.618	2.055	0.825	0.866	0.274	0.485	5.903	4.235
Motor Regulation	MR1	1.000	1	0.498	0.387	0.000	0.000		
	MR2	1.818	2.399	0.906	0.928	0.281	0.507	6.471	4.729
	MR3	1.744	2.151	0.869	0.832	0.285	0.485	6.131	4.435
	MR4	1.496	1.932	0.745	0.748	0.254	0.438	5.885	4.412
Delay Inhibition	DI1	1.000	1	0.737	0.589	0.000	0.000		
	DI2	0.890	0.664	0.656	0.585	0.159	0.113	5.611	5.871
	DI3	0.655	0.660	0.483	0.685	0.136	0.092	4.817	7.179
	DI4	1.051	0.773	0.775	0.693	0.203	0.110	5.165	7.056
Working Memory	WM1	1.000	1.000	0.518	0.579	0.000	0.000		
	WM2	1.381	0.835	0.715	0.637	0.250	0.166	5.529	5.023
	WM3	1.152	0.919	0.597	0.016	0.213	0.168	5.404	5.483
	WM4	0.370	0.023	0.192	0.422	0.193	0.153	1.912	0.147
Instruction Following	IF1	1.000	1.000	0.422	0.660	0.000	0.000		
	IF2	1.320	1.097	0.557	0.724	0.359	0.137	3.678	7.989
	IF3	1.832	1.225	0.773	0.808	0.496	0.172	3.692	7.127
	IF4	1.033	0.790	0.436	0.522	0.351	0.136	2.944	5.810
Attention Towards Owner	AO1	1.000	1.000	0.746	0.918	0.000	0.000		
	AO2	1.118	0.893	0.834	0.819	0.112	0.079	10.010	11.323
	AO3	1.090	0.820	0.813	0.753	0.085	0.075	12.861	10.912

**Table 3 animals-13-00533-t003:** Generalised additive model results for DEFS subscales. Significant *p*-values are marked in bold.

	Behavioural Flexibility			Motor Regulation			Delay Inhibition		
Predictors	Estimates	CI	*p*	Estimates	CI	*p*	Estimates	CI	*p*
(Intercept)	3.99	3.82–4.17	**<0.001**	3.08	2.89–3.28	**<0.001**	3.49	3.33–3.66	**<0.001**
Working status: No	*Reference*								
Working status: Yes	1.09	1.04–1.15	**0.001**	1.14	1.06–1.22	**<0.001**	1.09	1.04–1.16	**0.001**
Sex (Female/intact)	*Reference*								
Sex (Female/spayed)	0.96	0.91–1.01	0.102	0.91	0.85–0.98	**0.011**	1.03	0.97–1.08	0.344
Sex (Male/desexed)	0.93	0.89–0.98	**0.005**	0.84	0.78–0.90	**<0.001**	1.01	0.96–1.07	0.593
Sex (Male/intact)	0.98	0.93–1.04	0.504	0.91	0.84–0.99	**0.022**	1.00	0.95–1.07	0.874
Smooth term (Years)			0.672			**<0.001**			**<0.001**
Smooth term (Training Score)			**0.011**			0.486			0.487
Tensor product interaction (Years, Training Score)			0.519			0.162			0.122
Smooth term (Training Score) xWorking status: No			0.256			**<0.001**			**0.019**
Smooth term (Training Score) × Working status: Yes			0.820			0.916			0.519
Observations	954			954			954		
R^2^	0.032			0.152			0.031		
Deviance	3.03%			14.6%			2.98%		
	**Working Memory**			**Instruction Following**			**Attention Towards Owner**		
**Predictors**	**Estimates**	**CI**	** *p* **	**Estimates**	**CI**	** *p* **	**Estimates**	**CI**	** *p* **
(Intercept)	3.55	3.42–3.69	**<0.001**	3.71	3.57–3.85	**<0.001**	4.03	3.89–4.16	**<0.001**
Working status: No	*Reference*								
Working status: Yes	1.10	1.05–1.14	**<0.001**	1.11	1.07–1.16	**<0.001**	1.07	1.03–1.11	**0.001**
Sex (Female/intact)	*Reference*								
Sex (Female/spayed)	0.99	0.94–1.03	0.506	1.04	1.00–1.08	0.062	1.00	0.96–1.04	0.944
Sex (Male/desexed)	0.96	0.92–1.00	**0.044**	1.04	1.00–1.08	0.083	0.98	0.95–1.02	0.353
Sex (Male/intact)	1.02	0.98–1.07	0.317	1.00	0.95–1.05	0.943	0.98	0.94–1.03	0.462
Smooth term (Years)			**<0.001**			0.924			**<0.001**
Smooth term (Training Score)			0.178			0.708			**0.011**
Tensor product interaction (Years, Training Score)			0.075			0.482			0.127
Smooth term (Training Score) × Working status: No			**<0.001**			**<0.001**			**0.047**
Smooth term (Training Score) × Working status: Yes			0.581			0.723			0.950
Observations	954			954			954		
R^2^	0.082			0.157			0.246		
Deviance	8.07%			13.1%			24.7%		

**Table 4 animals-13-00533-t004:** *p*-values of pairwise Wilcoxon tests between different sex/reproduction status groups. *p*-values printed in bold are significant.

	Behavioural Flexibility	Motor Regulation	Working Memory
Female spayed/Female intact	0.0743	0.64515	0.551
Female spayed/Male desexed	0.0774	**0.00046**	0.137
Female spayed/Male intact	0.2216	**0.02176**	0.386
Female intact/Male desexed	**0.0041**	0.11774	0.496
Female intact/Male intact	0.3591	0.23522	0.298
Male desexed/Male intact	**0.0111**	0.79451	**0.047**

## Data Availability

The data presented in this study are available on request from the corresponding author. The data are not publicly available due to ethics approval conditions.

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
