# Peer review of "Executive Functions as Measured by the Dog Executive Function Scale (DEFS) over the Lifespan of Dogs"

_animals, 2023, doi:10.3390/ani13030533_

Round 1

Reviewer 1 Report

This is a useful research, Authors showed that the earlier developed Dog Executive Function Scale (DEFS) results in the same factor structure for juvenile and senior dogs as it did for 1-8 years old adult dogs in their original study. The paper has a solid statistical analysis section and the results are well described. I found the Discussion perhaps a little bit too lengthy, but otherwise this is a fine paper, I made only a very few comments on minor issues.

The Introduction I very well-written, it is concise but covers the relevant information about the assessment of executive functions both in humans and dogs.

What I wish to see in more detail though, it is a more sound reasoning why did the authors expect that the DEFS system may not be able to cover EFs of young and aged dogs as well? They show the literature that found that some elements of the EF complex show age-related differences, but this does not mean that the assessing instrument they developed would not able to catch these differences in the young, adult and aged dogs as well. In the Discussion I read some well-structured thoughts on this, however, it would be good to see these at the front end of the paper.

In case of Methods, some information would be good to read about what was considered as a ‘working dog’? What was necessary for achieving the ‘working status’? Am I guessing right that juvenile dogs could hardly get to the category ‘working dog’? Also, is there a chance that ‘working dogs’ were mostly purebreds? These questions would possibly good to be addressed at least in the Discussion.

Reviewer 2 Report

Review of manuscript id: animals-2135182, Foraita, Howell, & Bennett’s Executive Functions as measured by the Dog Executive Function Scale (DEFS) over the lifespan of dogs .

The purpose of this manuscript is to test the validity of the DEFS in younger and older dogs. As a secondary goal, Foraita et al. test whether the factors of executive functions are stable across different ages of dogs. I believe that the primary goal is of marginal importance but the secondary goal provides practical and theoretically important information about how the executive functions of dogs might change as the dogs age. As such, I believe that the manuscript will be of interest to the readers of the journal. Additionally, the manuscript is well written with minimal changes in style needed. My recommendation is that the authors revise and re-submit the manuscript.

My main concerns are:

Lines 181-183: The sample size for juvenile dogs (n = 129) and senior dogs (n = 127) seem a little small for a factor analysis with this many factors. Being what they are, some rules of thumb want 300 or more participants (Comrey & Lee, 1992; Tabachnick & Fidell, 2001) or 20 participants per factor (Arrindel & van der Ende, 1985). Other rules of thumb suggest that your sample size might be fine. While I do not advocate that the authors collect more data, they should cite appropriate literature to support their sample size.

Lines 184-194: Describe the sex and reproductive status of the dogs. If you collected breed information, describe that too. This information might help deal with potential confounds that are present in cross-sectional designs.

Line 316: “the Dog Executive Function Scale can be used with dogs as young as 3 months…” is not necessarily supported by your data. Your data suggest that in a group of dogs with ages from 3 to 11 months, the model is supported. But that may, or may not, be true for a given age especially if there were very few dogs with an age of 3 months in the data. It might be the case that the model fits very well for dogs aged, say, 5 to 11 months (perhaps the preponderance of your data) but that younger dogs do not have such a great fit.

Lines 422 and 434: It would be more accurate to say that the DEFS can be used to assess executive function in juvenile to senior dogs. Given the previous comment, it would be less appropriate to say 3 months to 15 years. Saying 0 to 15 years is not consistent with the evidence provided

In general, the authors conclude that age is the factor that influences some dimensions of the DEFS. However, this is a cross-sectional design – each dog is observed by its owner at one particular age. Such designs inherently are confounded. This limitation needs to addressed in the discussion.

Minor issues:

Line 25: Change “>1” (greater than 1) to “<1” (less than 1) and “<8” (less than 8) to “>8” (greater than 8).

Lines 142, 145, 146, Table 1: Superscript the square in χ².

Line 190: Delete the duplicate SD

Line 425: Change “Pet owner” to “Pet owners”.

Arrindell, W. A., & Van der Ende, J. (1985). An empirical test of the utility of the observations-to-variables ratio in factor and components analysis. Applied Psychological Measurement, 9(2), 165-178.

Comrey, A. L., & Lee, H. B. (1992). Interpretation and application of factor analytic results. Comrey AL, Lee HB. A first course in factor analysis, 2.

Tabachnick, B. G., & Fidell, L. S. (2001). Using multivariate statistics (4th ed.). Needham, MA: Allyn & Bacon.
